# KMS as a Sustainability Strategy during a Pandemic

**George Maramba \*** [ID], **Hanlie Smuts** [ID], **Funmi Adebesin** [ID], **Marie Hattingh and Tendani Mawela** [ID]

Department of Informatics, University of Pretoria, Pretoria 0028, South Africa; hanlie.smuts@up.ac.za (H.S.); funmi.adebesin@up.ac.za (F.A.); marie.hattingh@up.ac.za (M.H.); tendani.mawela@up.ac.za (T.M.)
\* Correspondence: u19083051@tuks.co.za; Tel.: +27-83-559-9597

**Abstract:** The 21st century world never anticipated a scenario in which it would be thrown into disarray by a fast-spreading viral disease, during which governments hastily had to enforce curfews by imposing travel and social gathering restrictions in order to contain it. The coronavirus disease of 2019 disrupted global supply chains and economies and caused death in every part of the world. Health departments and hospitals became the centres of attention as healthcare workers battled to save the lives of the infected. Governments struggled to calm citizens as the spread of incorrect and, sometimes, malicious information dominated all social media channels. The absence of established knowledge-sharing strategies and channels, knowledge about the disease or how to deal with the pandemic exacerbated the situation. This study investigates knowledge management systems as a sustainability strategy during a pandemic from three perspectives: understanding the disease, sourcing the required drugs and communicating with the citizens during a pandemic. The researchers adopted a survey research strategy for the study. The study makes an essential contribution to the value of KMS and the need to adopt them in the healthcare sector, particularly when faced with pandemics such as COVID-19.

**Keywords:** knowledge management systems; supply chain models; electronic supply chain systems; sustainable knowledge management systems; eHealth; e-Solutions during a pandemic

## 1. Introduction

The coronavirus disease 2019 (COVID-19) was discovered in Wuhan, Hubei province, in the People's Republic of China, in December 2019 [1]. It was declared a pandemic by the World Health Organisation (WHO) on the 11th of March 2020. The government of South Africa implemented the Disaster Management Act: Declaration of a National State of Disaster: COVID-19 (coronavirus) on the 15th of March 2020 [2]. On the 23rd of March 2020, the government enacted a lockdown, which took effect on the 26th of March 2020 [3]. In his press statement, the President of South Africa, Mr Cyril Ramaphosa, reiterated the need to avoid the spread of misinformation as it had a negative impact on the rest of the population by creating anxiety, stress and distrust towards the healthcare system and the government [3]. The lack of knowledge sharing and information contributed to conflicting messages and misinformation, which also lacked consistency and was generally of very poor quality [4,5].

Information and knowledge play a critical role in all aspects of life; in a chaotic and uncertain environment, it can mean the difference between life and death [1]. The management of knowledge management systems (KMS) is a critical element in which all modern organisations should invest resources [6]. A KMS entails the multidisciplinary collaboration of professionals who create rich knowledge repositories [1]. It is a set of organisationally defined processes, procedures, activities and knowledge used to implement knowledge management (KM) principles [6]. Healthcare organisations are crucial in all people's lives; therefore, the adoption of a KMS as a sustainability strategy in these organisations should be considered fundamental and mandatory [7]. The sharing of timely and accurate knowledge can save lives [8].

KMS enable organisations to make quick decisions, have consistent knowledge, reuse institutional and individual experiences to solve known problems, stimulate innovation and learn new insights based on acquired knowledge [9]. In addition, it provides the organisation with an opportunity to retain tacit knowledge, maintain proper content governance, authenticate knowledge and increase the focus on an outcome, which enforces best practices and works more effectively through knowledge refinement [6,10]. A pandemic calls for multidisciplinary healthcare teams to collaborate and work together with non-medical entities [5]. COVID-19 exposed a disconnect between citizens and their governments [5]. Furthermore, at that time, government departments were communicating conflicting messages to citizens [4], creating distrust and civil disobedience.

The inconsistent communications from governments to citizens through healthcare departments and other healthcare institutions created a rift between citizens and their governments [5,11]. Hence, the guiding research question for this study is: What role do knowledge management systems play during a pandemic or disaster? The main objective of this study is to determine if there were deficiencies during the COVID-19 pandemic that could have been better managed if there had been effective KMS and how such systems could have improved the situation, further determining if KMS can be a sustainable strategy to mitigate a pandemic.

## 2. Background

The major challenge that healthcare institutions faced in the initial phases of the pandemic was defining new logistics and supply chains, as necessitated by the imposed curfews and travel and social gathering restrictions [1,5,8]. At the peak of the COVID-19 pandemic, there were disagreements in many countries over the distribution and production of supply chains due to industrial shutdowns [8], a lack of medical supplies and unequal distribution of equipment, such as ventilators [5]. Most governments assumed control of sourcing COVID-19 supplies, such as vaccines and ventilators; however, this created more chaos in the initial phases because of inadequate coordination, miscommunication and inconsistent knowledge in various government departments [12,13]. These shortcomings caused anxiety and mistrust between governments and citizens, and the absence of proper communication and information created an opportunity for citizens to speculate, resulting in a lot of misleading and counter information [5]. Consequently, citizens panicked, and a sporadic spread of misinformation about the pandemic occurred, which created unnecessary anxiety and pandemonium in many communities.

The pandemic further placed the remaining supply chain systems under immense pressure [9]. In addition, healthcare practitioners became stressed and overwhelmed [1,10]. When knowledge is managed and distributed properly, it can help manage a pandemic crisis and calm the citizenry [10]. During the COVID-19 pandemic, logistics and supply chain systems lacked the aspects of a KMS, which is a valuable and sustainable strategy wherever coordination, collaboration and knowledge are required [14], particularly during a pandemic.

During the pandemic, healthcare workers operated under immense pressure as they had to attend to almost everything within their environment without the requisite knowledge [12]. The adoption and use of KMS would have enabled governments to contain infection and reduce exposure to risk before COVID-19 became a pandemic [11,15]. The COVID-19 pandemic exposed a disconnect between healthcare organisations and departments of health [11,13], resulting in conflicting approaches to combating and addressing the pandemic. The healthcare practitioners' differing views about the pandemic created uncertainty because some of these views found their way into social media and were further manipulated, causing counter-knowledge [16].

In the first three to four months of the pandemic, healthcare practitioners did not have adequate knowledge of how to treat the virus [17]. There were many conflicting views and opinions on how to treat the disease while the virus was spreading rapidly to all parts of the world [18]. The dominant benefit of adopting a KMS is the creation of

defined and trusted sources of single truths or facts [18], thereby creating well-structured repositories that enable easy referencing and collaboration [9,13]. The lack of information and knowledge structures exposed by the pandemic [19] could have been avoided if KMS were already entrenched in healthcare institutions and federal governments.

Most governments failed to mitigate fear, panic and misinformation among their citizens, which opened space for the creation of erroneous perceptions and non-factual conspiracy theories [11]. There was much counter-knowledge from unverified sources, such as hoaxes, rumours and lies about COVID-19 [11,16]. Counter-knowledge spreads faster than facts and correct information. As a result, healthcare authorities were constantly battling to mitigate and dismiss such counter-knowledge. At times, governments did not have the knowledge or factual information to substantiate or dismiss lies [11], while, in other instances, healthcare authorities took too long to respond. This situation was exacerbated by the absence of KMS [8,19]; there were moments when it seemed as if none knew what had befallen the world. Medical scientists were confronted by individuals, groups and companies who contested the official truth, proposing unsupported solutions [16], which, moreover, happened on public platforms, creating unnecessary confusion. Individuals' knowledge bases are created from what is regarded as official truth as well as rumours and unsupported information, causing distrust of official truth or knowledge [16]. The pandemic could have been managed much better if KMS were in place and had been adopted, both before and during the pandemic [8,11,16]. In their studies, the scholars Ammirato, Linzalone [9]; Chaturvedi, Singh [13]; Bolisani, Cegarra Navarro [16]; and Pinto [1] highlighted the need for the urgent adoption of KMS to manage any future pandemics more effectively.

## 3. Benefits of Knowledge Management Systems during a Pandemic

The benefits of using a KMS during a pandemic are presented from four perspectives, namely healthcare practitioners, citizens, manufacturers of vaccines and medical drugs and the Department of Health.

Healthcare practitioners

- An updated KMS provides knowledge at any time, removing human dependencies [20], which is ideal in a healthcare institution during a pandemic.
- Mid-crisis during the COVID-19 pandemic, many critical medical practitioners died, resulting in healthcare knowledge loss [11]. A KMS would enable healthcare institutions to retain any knowledge gained over time [20].
- A KMS provides an unlimited environment for rapid collaboration and enrichment of the captured knowledge, which could speed up the containment of a pandemic [15].
- A KMS enables collaborators and scientists to find a solution much faster since it provides a lot of information, which can be used to gain insights [21].
- A KMS provides healthcare practitioners with opportunities to detect fast-spreading diseases and trace their origin before such diseases become pandemics [22].
- Disease profiling and trend analysis can easily be drawn from a KMS, thereby enabling practitioners to generalise remedies and standardise treatment procedures [23].
- A KMS enriches healthcare practitioners' skills and knowledge about disease characteristics, further capacitating them with appropriate and tested medicine and drug knowledge [24].

Citizens

- A KMS enables citizens to receive treatments using the best-known procedures and the ideal medication that has been reviewed by many healthcare experts [23].
- KMS reduce incorrect diagnoses because the relevant knowledge is already profiled [7].
- Citizens receive improved healthcare services as a KMS provides an opportunity for innovation [24].
- Citizens receive consistent knowledge of diseases and medical drugs, and the KMS removes subjective and personal opinions, moving towards a scientific position [23].

Manufacturers of vaccines and medical drugs

- Manufacturers have a platform to coordinate with their stakeholders using a KMS-based supply chain [25].
- A KMS enables manufacturers to streamline their resources and improve production by simulating and changing production variables to determine their optimum production levels for mitigating a pandemic [1,26].
- A KMS enables organisations to find effective and efficient ways to run production operations [26].
- Organisations can create and embed knowledge in KMS-driven processes and procedures [1].

Department of Health

- A KMS enables the government to communicate consistent information to citizens, which is necessary for citizens to cooperate with the government [11].
- At times during the early months of the pandemic, governments were not reaching out to all healthcare stakeholders [5]. Conversely, a KMS enables the Department of Health to identify all the stakeholders and collaborate with all involved.
- By eradicating counter-knowledge during a pandemic [11], a KMS could provide the government with adequate knowledge and information to dismiss the malicious "myths" that flood social media.
- Most governments encountered vaccination apathy [5,11] caused by counter-knowledge and governments' conflicting statements going into the public space unchecked and uncontested.

## 4. Methodology

A survey research strategy was adopted for this study as it enabled the researchers to review several KMS capabilities during a pandemic. The researchers applied a document review technique to analyse the data for this study while exploring the application simulation to evaluate the supply chain system. Three potential KMS areas were identified based on the benefits of the critical areas discussed in the previous section of this article. Each of the identified KMS areas was evaluated with regard to its role and functionality, namely supply chain, disease profiling and treatment and a department of health. The study explored these three areas as separate business cases to determine if the pandemic would have been managed better and if KMS can be considered as a sustainability strategy during a pandemic. The next sections discuss the three cases.

Business Case 1: The first case was explored to select supply chain management software that could enhance decision-making during a pandemic. The objective of this business case was to enable the researchers to determine if a KMS-driven supply chain could aid and speed up decision-making by providing predictions. The software advice website (Available online: https://www.softwareadvice.com/ (accessed on 19 August 2022)) is a business database engine designed for software evaluations and was used to identify the top five supply chain applications for 2022 used in the healthcare sector. Licensed users who had purchased and used the software provided the software application ratings. The keywords "supply chain management software in healthcare" were used to perform the search, which returned 170 applications. The researchers downloaded the trial versions of the top five applications, including their user manuals, which are presented in Table 1.

The researchers gave all the applications adequate testing time based on their user manuals. The Anylogistix application (The Anylogic Company) was selected based on three major aspects, namely functionality, pricing models, support of digital twin and predictions.

**Table 1.** The year 2022 top five supply chain management software in healthcare.

| Software Name | Rating | Website |
|---|---|---|
| e-Procure | 5.00 | https://get.e-procure.net/, accessed on 17 January 2023 |
| Precoro | 4.80 | https://get.precoro.com/, accessed on 17 January 2023 |
| Anylogistix | 4.47 | https://www.anylogistix.com/, accessed on 18 January 2023 |
| Route4Me | 4.42 | https://www.route4me.com/, accessed on 18 January 2023 |
| Odoo | 4.10 | https://www.odoo.com/, accessed on 17 January 2023 |

Business Case 2: The second case was conducted to establish a disease profiling application that would be ideal for profiling diseases such as COVID-19. The world's two most popular KMS web applications known for disease profiling are the Centres for Disease Control and Prevention (CDC, https://www.cdc.gov/, accessed on 20 February 2023) and the Mayo Clinic (https://www.mayoclinic.org/, accessed on 21 January 2023). The CDC is designed for disease profiling and prevention, whereas AskMayoExpert is designed for disease profiling and to provide medical practitioners with the requisite knowledge to treat a disease, after which it records the practitioners' feedback into the system to enrich the knowledge. Furthermore, this application is integrated with the healthcare system. The researchers deliberated to determine the most ideal of these two knowledge bases and opted for the AskMayoExpert (Mayo Clinic knowledge base).

Business Case 3: This business case was conducted to establish if KMS could have enabled a government to communicate consistent information to the citizen, eliminating conflicting statements. This study was conducted in South Africa; the researchers explored the websites that were put in place by the South African Department of Health to communicate information about COVID-19. The researchers identified two websites complying with the keywords "South African Department of Health" and "COVID", namely https://sacoronavirus.co.za (accessed on 26 February 2023) and https://www.gov.za/Coronavirus (accessed on 26 February 2023). These two websites were both considered for evaluation and content analysis.

## 5. Results

### 5.1. Supply Chain-Anylogistix Software

The COVID-19 pandemic exposed the shortcomings in the current logistics and supply chain systems in a vast number of industries, including the healthcare sector [4,27]. It took a lot of effort and time for organisations to reorganise and become operational, as they had not anticipated the magnitude of the disruption caused by the pandemic [8]. The distribution of vaccines and protective clothing is the top priority during a pandemic for saving lives [19]. A checklist was prepared to determine the aspects of an effective KMS supply chain during a pandemic. The supply chain checklist is presented in Table 2, which depicts each feature with its summary.

**Table 2.** A KMS supply chain checklist.

| Feature | Summary |
|---|---|
| Ease of deployment | The KMS must be easy to deploy across many platforms [15]. |
| Scaleability | It should allow for upscaling or downscaling [28]. |
| Integrate with other applications | Integration with other applications enables interactive collaboration [15]. |
| Track inventory and distribution chain | A KMS that provides inventory management and distribution capability is critical during a pandemic [4]. |

**Table 2.** *Cont.*

| Feature | Summary |
|---|---|
| Manage suppliers and raw materials | Curfews and travel restrictions were disruptive during the pandemic, and knowledge about the suppliers of raw materials plays an important role in keeping production running [10]. |
| Manage orders and demand | An automated KMS supply chain model that predicts demand based on the escalation of the pandemic can aid decision-makers [1]. |
| Manage production | The production of vaccines for the whole world is an enormous task, particularly considering third-world countries that do not have production facilities [18]. Therefore, a KMS that enables the management of production could enable drug manufacturing companies to create small production plants anywhere in the world. |
| Manage transportation | The distribution of vaccines requires proper planning of transport logistics from cargo airliners to vaccination sites. Therefore, a KMS with this aspect could ease the burden of logistics planning [1]. |
| Ability to determine costs | Production costs, transport costs and operating costs are critical aspects to keep sight of during a pandemic so that adequate funding is sourced and made available [13]. |
| Time | Time statistics are an important aspect during a pandemic, as the time taken to manufacture vaccines, the time taken to ship and the time vaccines should stay in transit should be known in advance to enable transportation planning [11,29]. |
| Performance indicators and statistics | Performance indicators provide red and green flags in the production line and supply chain, as well as in operations. These are essential elements during a pandemic as they could enable the improvement of processes and also identify bottlenecks [30,31]. |
| Visualisation and digital twin simulation | With data and modern technology, it is now possible to visualise and simulate a real-world scenario [4]; a KMS with these options will enable the simulation of real-world scenarios. |
| Financial statistics | Drawing financials from a KMS will enable an understanding of true values from the supply chain. |
| Availability and failover cluster | The supply chain system must work even if it is offline; it should have automatic recovery functionality and failover clusters [30,31]. |

The features identified in Table 2 will enable the healthcare sector to eradicate potential supply chain challenges. In addition, data analysis, reporting and simulation were considered wherever available. The researchers selected Anylogistix software version 2.15.3.202209061204 based on the above features. It is an integrated and advanced KMS supply chain package that addresses most of the identified aspects in Table 2, including the generation of "digital twins", which can be simulated to reflect real-world scenarios.

Figure 1 depicts a visual simulation of vaccination sites in KwaZulu-Natal Province in South Africa (blue), distribution centres (red), suppliers (green), provincial warehouses (brown) and the road network. The figure presents the projected figures for vaccines delivered per month, calculated based on the province's population (11.5 million).

The software's licencing and easy migration of licenses make it easy to dynamically shift and change resources during a pandemic. It had the most affordable five-user license compared to the other four applications being considered. The research team proceeded to purchase the software, which took less than 10 min online (including the licensing). In addition, a software technician was allocated to familiarise the team with the fundamentals.

Upon using the Anylogistix software, the researchers were able to predict the vaccines required for the province, the number of vaccinations per day, the rate of COVID-19 infections, the requisite number of medical staff and many insights and vital knowledge accurately. The software further offered provisions for managerial decision-making. In addition, the software enabled interactive modelling and the simulation of real-world scenarios. A supply chain embedded within a KMS is critical during a pandemic as it presents decision-makers with visuals [30] of essential medical sourcing, logistics, quantities, costs overview, population distribution and drop points. The Anylogistix software proved to be

an effective knowledge repository; it stored detailed supply chain data and information that could be used for advanced analytics and simulation. During a pandemic, decision-making needs to be fast, correct and well-informed; thus, an integrated supply chain with a KMS will save lives and expedite the eradication of a pandemic.

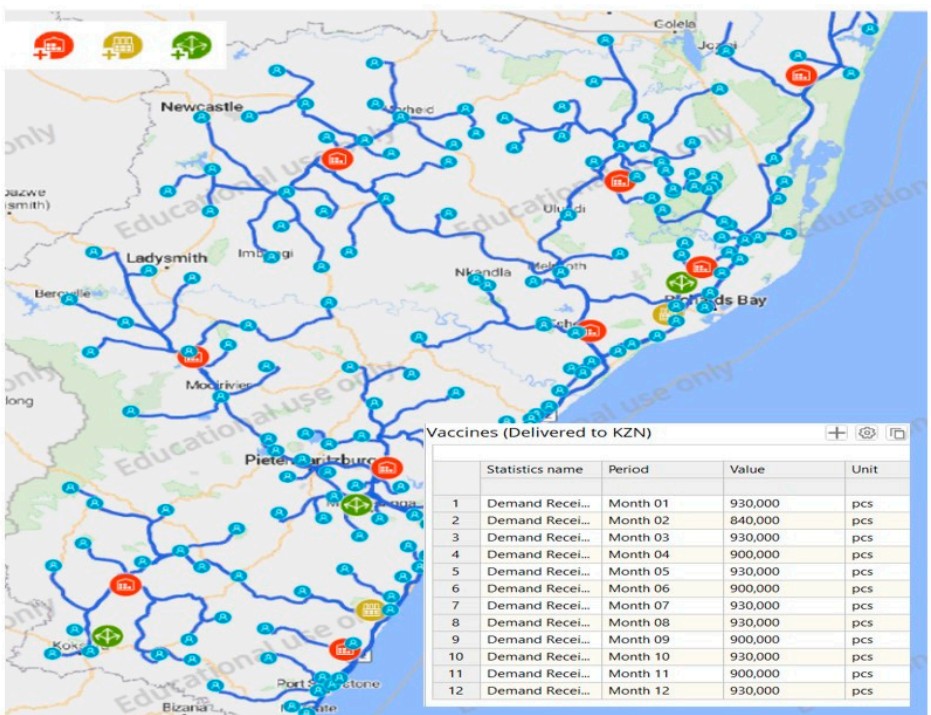

**Figure 1.** KwaZulu-Natal Anylogistix simulation.

*5.2. Disease Profiling and Treatment–AskMayoExpert*

The COVID-19 variants were invasive during the peak of the pandemic as there was no adequate information and knowledge available [23,32]. It became apparent that without a disease knowledge base, it took longer to manufacture the drugs required to combat the pandemic and start drug trials during a pandemic while people could be dying [11]. Some critical aspects were considered to obtain an effective, reliable and authoritative KMS that would be ideal during a pandemic, as presented in Table 3.

**Table 3.** Disease knowledge base KMS.

| Feature | Summary |
|---|---|
| Scientific collaboration | A comprehensive and authoritative KMS should allow for scientists' input and collaboration and also validate the knowledge acquired. |
| Medical specialists' collaboration | Medical specialists play an important role since they overlap between medical practice and scientists; their contribution is significant to a KMS during and after a pandemic. |
| Medical practitioners' collaboration | Medical practitioners require knowledge for their practices—they treat patients and are on the front line of identifying the effects of drugs and medical procedures. Their feedback on KMS is crucial since it creates and authenticates procedures and drugs. |
| Disease profile | Disease profiling is essential, particularly during a pandemic. As such, COVID-19 exposed the healthcare sector to an environment with diffused knowledge—most countries did not have disease knowledge bases [21]. |

**Table 3.** *Cont.*

| Feature | Summary |
| --- | --- |
| Medical drugs | Adequate knowledge about drugs, such as their side effects, quantities and background on where they have been used or tested, is important because it enables medical practitioners to work faster and more effectively using approved drugs during a pandemic [11]. |
| Professional recommendations | Other medical professionals in the healthcare sector also play significant roles; their contributions and collaborations in KMS formation and refinement can make a difference during a pandemic. |
| Research and development | Innovations, improvements and continuous research are essential aspects during and after a pandemic, and the lessons learnt shape the future [15], i.e., a KMS is a platform for hosting this type of knowledge. |
| Client interaction | A KMS must allow some interaction with clients, particularly the patients; they do not need to view detailed and technical information but rather disease-profiled data. |
| Disease trend analysis | A KMS with detailed information provides a platform for performing trend analysis to infer the patterns of diseases [13]. |
| Irregularity triggers | Events and data-based triggers provide warnings to knowledge users and experts. A KMS with these types of triggers provides timeous warnings, i.e., before a pandemic takes effect. Such a KMS allows healthcare sectors and governments to plan for a pandemic. |

Figure 2 presents a snapshot of the AskMayoExpert KMS showing the extraction of the COVID-19 knowledge page. It was extracted from the Mayo Clinic International website (Available online: https://ce.mayo.edu/content/askmayoexpert (accessed on 21 August 2022)). The knowledge base presents information in a variety of formats, such as text, recorded videos and simulated interactive hypertext, to meet different audiences' needs.

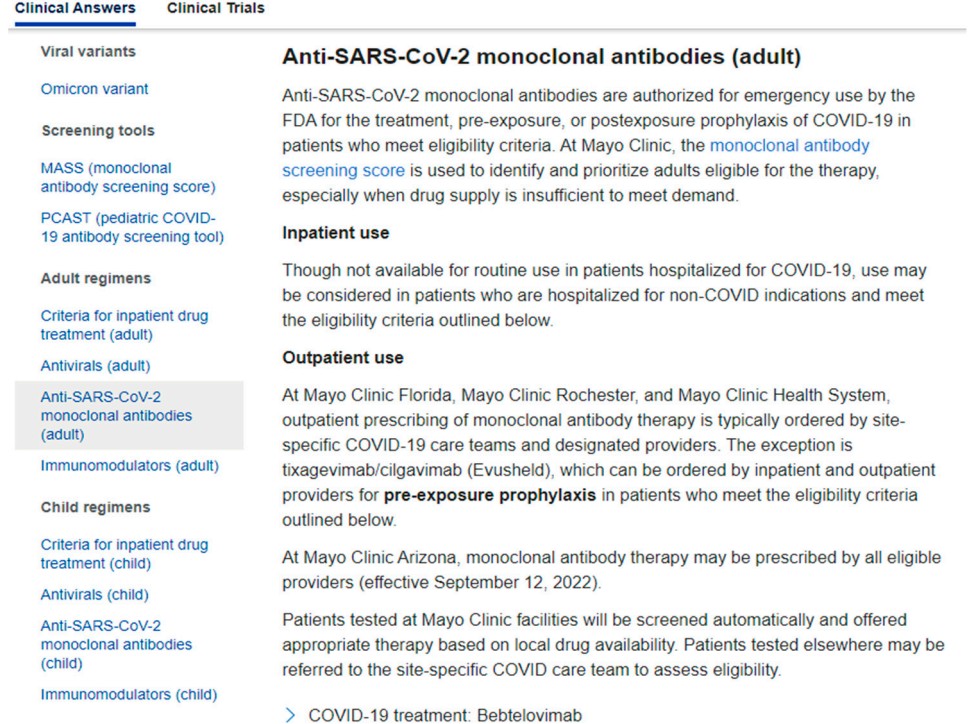

**Figure 2.** AskMayoExpert COVID-19 knowledge base. [Available online: https://ce.mayo.edu/content/askmayoexpert, accessed on 3 January 2023].

The AskMayoExpert KMS provides a platform for knowledge creation and reviews to medical specialists, practitioners, scientists and patients. The researchers chose AskMay-

oExpert because it facilitates collaborations and reviews by all healthcare participants. Figure 2 shows the COVID-19 knowledge extract. The extract is very informative, both to non-medical and medical staff, and there is adequate knowledge for a medical practitioner to treat patients without wasting time. A KMS on diseases would save lives during a pandemic by speeding up the treatment process, reducing incorrect diagnoses and enabling innovation, in-depth studies and investigation on a pandemic using collected data.

### 5.3. Department of Health-Websites

After identifying a supply chain and disease profiling KMS in Business Cases 1 and 2, the researchers investigated the KMS provided by the Department of Health in South Africa.

From a KMS perspective, the two websites provided different sources of information and knowledge. The main focus of the website in Figure 3 was to present daily and weekly COVID-19 cases, such as the number of vaccinations, the number of deaths, the recovery rates and the vaccines administered. The website (Figure 3) was dedicated to the COVID-19 pandemic and only contained information and data relevant to the topic. The second website, depicted in Figure 4, focused on management information (e.g., transitional measures), legislation, definitions of the different lockdown levels (alert levels), travel restrictions, home safety tips and organisational safety guidelines concerning COVID-19. This website (Figure 4) was not dedicated to the pandemic only as it contained information about other health-related services provided by the Department of Health. Both sites provided the standard website search and site map capabilities. The website in Figure 3 utilised a revolving banner to communicate multiple aspects regarding the pandemic and made it difficult to find the information the researchers were looking for quickly since it displayed too much in an unstructured manner.

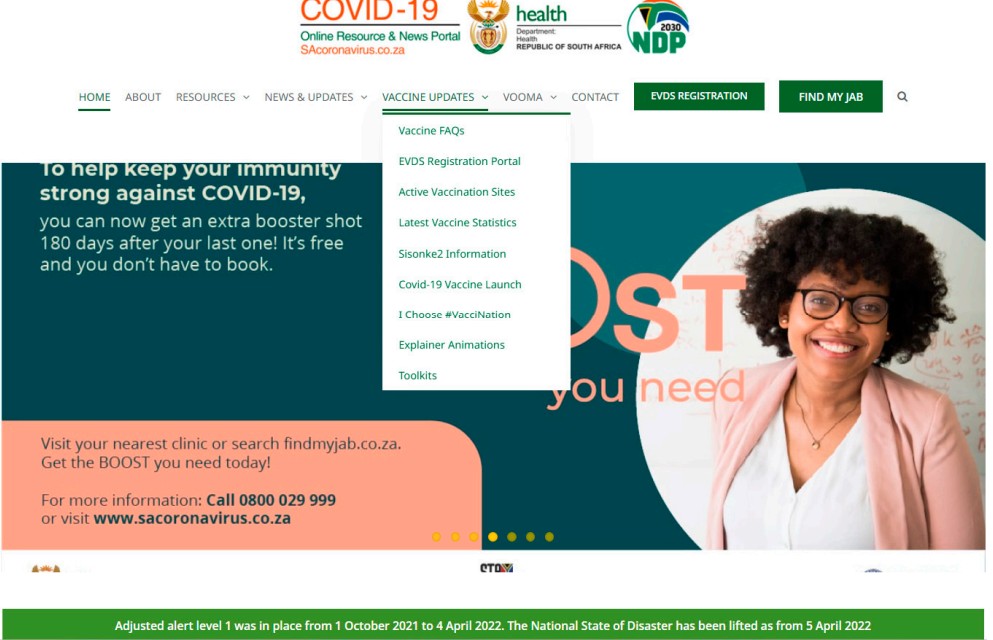

**Figure 3.** Website 1 (https://sacoronavirus.co.za/, accessed on 26 February 2023).

The two websites contributed to short- and long-term measures, leveraging both internal and external knowledge resources. Furthermore, they supported knowledge exploration, demonstrating the deployment of emergent knowledge management strategies.

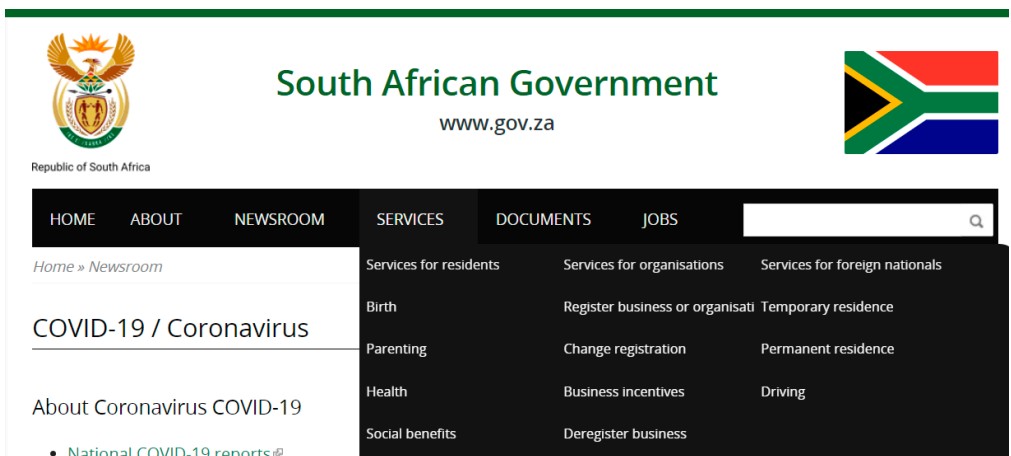

**Figure 4.** Website 2 (https://www.gov.za/Coronavirus, accessed on 26 February 2023).

*5.4. KMS as a Sustainability Strategy during a Pandemic*

The identified business cases are vital and, therefore, require the adoption of a KMS to manage and eradicate a pandemic such as COVID-19. These include disease identification, supply chain models and communication. The three cases are summarised in Table 4, which depicts the sections, business cases and roles of a KMS during a pandemic.

**Table 4.** The role of a KMS during a pandemic.

| Section | Business Case | Role of the KMS during a Pandemic |
| --- | --- | --- |
| Identification of the pandemic | Disease profiling—AskMayoExpert | • Quick detection of fast-spreading diseases and tracing them to their points of origin.<br>• Provides disease trends, patterns and knowledge.<br>• Available information about the disease could enable scientists to commence working on remedies timeously without having to start fresh investigations.<br>• Eases the tracking of drug testing.<br>• Provides the capability to measure infection rates.<br>• Identifies affected populations and concentrates on them.<br>• The future reuse of the knowledge repository to gain more insights.<br>• Enables the development of standardised diagnosis.<br>• Required procedures can be generated and shared across users and can be further enhanced with more learning.<br>• Enables the optimisation of procedures and processes.<br>• Increases the chances of correct diagnosis and eradicates human errors.<br>• Eliminates knowledge loss when experienced practitioners retire or relocate.<br>• The healthcare sector is already overloaded with current challenges; thus, knowledge repositories have become a necessity for keeping and storing critical knowledge for future use and reference.<br>• The provision of innovation and continuous improvement as ways to combat the pandemic by performing replays using available information. |

**Table 4.** *Cont.*

| Section | Business Case | Role of the KMS during a Pandemic |
|---|---|---|
| Sourcing of drugs | Supply chain—Anylogistix application | • Facilitates interactive sourcing of raw materials.<br>• Managing drug production, demand and supply; providing adequate data analysis, trends, reporting and predicting to assist managers in making informed decisions.<br>• A supply chain integrated with a KMS logically connects those in need of the drugs to the manufacturers and raw materials, thereby providing a visual simulation of reality.<br>• The KMS supply chain model provides valuable knowledge and insights, such as transportation costs, production capacity and costs, the rate of drug utilisation and demand, a raw material supply network and distribution channels.<br>• Optimising the supply chain to combat the pandemic effectively.<br>• The KMS supply chain model can be easily scaled up or down to align with a specific situation.<br>• A KMS supply chain model can further provide health procedure automation using collected information and knowledge.<br>• Supply chain risk assessment and evaluation.<br>• Inventory analyses, trends and predictions. |
| Communication and administration of the population and resources | Communication—government websites | • KMS-based communication channels enable federal governments and their respective health and communication departments to present consistent and informed information without conflicting views.<br>• A KMS-integrated website should be a single point of reference; it must contain all the necessary information and not refer to other repositories or websites so as not to divert or redirect users to dead links.<br>• The KMS website has to be the only point of reference.<br>• A KMS enables the elimination of knowledge gaps.<br>• Enables defining pandemic response procedures, vaccination processes, communication protocols, help centres and information desks.<br>• The KMS should provide general questions and answers that might be circulating on social media to denounce all counter-knowledge and misinformation. |

Table 4 summarises the findings of this study; each phase is mapped to its respective business case, and the roles of the KMS are then mapped against each case. The identification of the pandemic is mapped to the disease profiling software, AskMayoExpert, and the sourcing of the drugs is mapped to the supply chain system, while communication and administration are mapped to the government websites. The role of a KMS during the pandemic's key elements is presented in Table 4 to enrich and reveal the value of embedding a KMS in these critical sections when dealing with a pandemic.

The three cases presented in Table 4 should be administered during and after a pandemic; the development of a disease profiling knowledge base, supply chain and pandemic administration platforms embedded within a KMS should be ongoing processes for them to remain sustainable strategies to eradicate a future pandemic, particularly after the lessons learnt with the COVID-19 pandemic. A KMS now becomes the ideal solution for combating a future pandemic; it should be applied towards shaping the healthcare sector and all the relevant organisations.

## 6. Future Research

KMS platforms are critical artefacts in the healthcare sector, particularly during a pandemic, as they equip medical practitioners with much-needed knowledge to act swiftly. There is a need for more studies to be conducted to create awareness of the value of KMS

in the healthcare sector so that organisations can assign more resources to their creation and support. This study could not explore or evaluate all the available knowledge systems.

## 7. Conclusions

This study explored the role of KMS during a pandemic and identified the benefit areas where they could have improved the situation. In addition, the study explored the KMS as a sustainability strategy to combat a pandemic; the adoption and implementation of a KMS are critical as they could help save lives during a pandemic. The business cases assisted the researchers in evaluating the role of the KMS from various perspectives, thereby identifying it as a sustainable strategy for managing a pandemic. Collaborating and sharing knowledge help refine processes and procedures and create effective products through innovative technological artefacts. The supply chain, understanding disease and coordinating the healthcare environment are some of the critical dimensions that will benefit from the adoption of a KMS during a pandemic. If vaccines were provided on time, medical practitioners would know how to treat and contain the disease, and while the government coordinated and communicated effectively with its citizens, the pandemic would not disrupt an entire country, nor would it create distrust and counter-knowledge.

**Author Contributions:** Conceptualization, F.A.; Validation, M.H.; Formal analysis, F.A.; Data curation, T.M.; Writing—review & editing, G.M.; Project administration, H.S. All authors have read and agreed to the published version of the manuscript.

**Funding:** The research reported in this article was supported by the South African Department of Science and Innovation (DSI) and the South African Medical Research Council (SAMRC) under BRICS JAF #2020/033. The content and findings reported or illustrated herein are solely the deductions, views and responsibility of the researcher/s and do not reflect the official positions and sentiments of the funders.

**Data Availability Statement:** The data used to perform supply chain simulation was obtained from the three authenticated websites namely, Statistics South Africa, National Institute for Communicable Diseases and the Department of Health COVID-19 official site. The websites are listed below; Population data: Statistics South Africa: https://www.statssa.gov.za/ accessed 5 June 2023. COVID-19 Statistical data: National Institute of Communicable Diseases: https://www.nicd.ac.za/wp-content/uploads/2021/ accessed 5 June 2023. Department of Health COVID-19 website: https://sacoronavirus.co.za/covid-19-daily-cases/ accessed 26 February 2023.

**Conflicts of Interest:** The authors declare no conflict of interest.

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
