# Peer review of "KMS as a Sustainability Strategy during a Pandemic"

_sustainability, doi:10.3390/su15129158_

Round 1

Reviewer 1 Report

Although the article reads well, it lacks a few elements typical of a scientific
study. First of all, it lacks research questions and hypotheses.
Therefore, it is difficult to verify whether the conducted proving allows to achieve
the goal, which the authors do not specify either.
So I suggest supplementing the text with these basic elements.
In my opinion, there was also a lack of discussion on the results, or rather recommendations resulting from the analysis of three different case studies

The summary also requires some changes, i.e. the answer to the question posed in the introductory part (as I have already mentioned, such question(s) are missing and should be supplemented)

minor linguistic corrections should be made

Author Response

Thank you for the feedback and review, kindly find our feedback with regard to the recommendations.

Point 1: Minor English corrections

Response: The paper has been revised, in addition, the paper has been language edited.

Point 2: Are the research design, questions, hypotheses and methods clearly stated?

Response: Thank you for highlighting this, we have added a guiding research question in the introduction section.

In addition, two objectives have been added to provide clarity regarding the direction of the study - this has been added in the introduction section.

Point 3: Are the conclusions thoroughly supported by the results presented in the article or referenced in secondary literature?

Response: The conclusion has been revised and aligned with the guiding research question and objectives identified in the introduction.

Reviewer 2 Report

1. In the abstract, too much background is presented, the authors should focus on the knowledge management systems and how they are applied to specific sustainability strategies during a pandemic.

2. The keywords, “Supply chain models” and “electronic supply chain systems” should be selected as one of them. More keywords should be selected.

3. In the introduction, the authors should give a more refined summarization of KMS and their advantages during Covid-19. “Hubel province” should be corrected as “Hubei Province”.

4. In the background, more references focusing on different KMS and their applications should be given and summarized.

5. In the methodology, the authors took some existing systems with websites as cases of KMS capabilities, which cannot be considered as the methodology.

6.  In the results, the authors should the functions and applications for different systems, however, what are the specific sustainability strategies?

Written English must be improved.

Author Response

Thank you for the feedback and review, kindly find our feedback with regard to the recommendations.

Point 1: In the abstract, too much background is presented, the authors should focus on the knowledge management systems and how they are applied to specific sustainability strategies during a pandemic.

Response: We have reviewed and rewritten the abstract

Point 2: The keywords, “Supply chain models” and “electronic supply chain systems” should be selected as one of them. More keywords should be selected.

Response: New keywords have been added as recommended, thank you. 

Point 3: In the introduction, the authors should give a more refined summarization of KMS and their advantages during Covid-19. “Hubel province” should be corrected as “Hubei Province”.

Response: The summarization of KMS and its advantages are already in paragraphs 2 and 3 of the introduction.

Hubei Province has been corrected, thank you. 

Point 4: In the background, more references focusing on different KMS and their applications should be given and summarized.

Response: Paragraph 4 has been added to the background section to address this aspect.

Point 5: In the methodology, the authors took some existing systems with websites as cases of KMS capabilities, which cannot be considered as the methodology.

Response: This section was enhanced, two surveys and an application simulation were conducted in this study.

Point 6: In the results, the authors should the functions and applications for different systems, however, what are the specific sustainability strategies?

Response: The section: "KMS as a sustainability strategy during a pandemic", we have added a new paragraph (last one) to address this concern

Point 7: Extensive editing of English language required

The study has been language edited

Reviewer 3 Report

English usage is appropriate (there are a few minor errors). 

Author Response

Thank you for the feedback and review, kindly find our feedback with regard to the recommendations.

Point 1: Language editing

Response: The study has been language edited

Point 2: Are the research design, questions, hypotheses and methods clearly stated?

Response: This has been addressed: added a guiding research question and objectives to provide more clarity of the study.

Point 3: Are the conclusions thoroughly supported by the results presented in the article or referenced in secondary literature?

Response: The conclusion has been reviewed to align with the guiding research question and objectives.

Round 2

Reviewer 1 Report

Research questions were supplemented, the goal was defined - the work gained value. I therefore have no further objections  

I have no comments - the editing process will probably reveal minor errors (typos, etc.), but in general the article is correct in terms of language  

Author Response

Thank you

Reviewer 2 Report

This manuscript has improvements. Some mistakes should be corrected, such as "KMSs" or "KMSes", etc. Please check the details of the manuscript.

Please check the spelling again.

Author Response

Thank you for the feedback and review, kindly find our feedback with regard to the recommendations.

Point 1: Some mistakes should be corrected, such as "KMSs" or "KMSes", etc. Please check the details of the manuscript.

Response: We have reviewed and removed the KMSs and KMSes, thank you

Point 2: Are the research design, questions, hypotheses and methods clearly stated?

Response: We have added an objective to each business case

Point 3: For empirical research, are the results clearly presented?

Response: Further elaborated business case 1 results discussion

Point 4: Please check the spelling again.

Response: Revised the manuscript, mistakes were picked up and corrected, thank you